# Quality of Diet of Patients with Coeliac Disease in Comparison to Healthy Children

**DOI:** 10.3390/children9101595

**Published:** 2022-10-21

**Authors:** Sara Sila, Mario Mašić, Draženka Kranjčec, Tena Niseteo, Lucija Marić, Ana Radunić, Iva Hojsak, Oleg Jadrešin, Zrinjka Mišak

**Affiliations:** 1Referral Centre for Pediatric Gastroenterology and Nutrition, Children’s Hospital Zagreb, 10000 Zagreb, Croatia; 2Health Center Zagreb East, 10000 Zagreb, Croatia; 3Croatian Academic Centre for Applied Nutritional Science, 10000 Zagreb, Croatia; 4School of Medicine, University of Zagreb, 10000 Zagreb, Croatia; 5School of Medicine, University J.J. Strossmayer Osijek, 31000 Osijek, Croatia

**Keywords:** coeliac disease, gluten-free diet, children, adolescents

## Abstract

A lifelong strict gluten-free diet is the only available treatment for patients with coeliac disease (CD). As with any restrictive diet, it may potentially lead to nutritional deficits. Seventy-six patients with CD (mean age 9.0 ± 4.3 years, 57% female) and 590 healthy controls (HC) (mean age 9.9 ± 0.1 years, 54% female) were recruited and requested to keep a 3-day food record (3DFR). In HC patients, anthropometric data were determined at the time when 3DFRs were collected. In CD patients, anthropometric data were determined at two time points: at diagnosis and at the time of 3DFRs collection. Intake of energy, macronutrients, and micronutrients was determined using PRODI expert 6.9 software and expressed as a percentage of recommended daily intake. In CD patients, all measured anthropometric measures (body weight (BW), body height (BH), and body mass index (BMI) z-scores) increased significantly after the mean duration of 34.1 months of a GFD. Overall, CD patients had significantly lower BW and BH z-scores compared to healthy controls. Patients with CD were generally more compliant with the recommended dietary intakes of macronutrients and some micronutrients, as compared to HC. Three participants were not compliant with the GFD; 42.1% of participants took oral nutritional supplements at the start of their GFD. Our study showed that patients with CD have better compliance with dietary recommendations compared to healthy controls, showing that a well-balanced GFD diet can provide necessary macro- and micronutrients.

## 1. Introduction

Coeliac disease (CD) is an immune-mediated systemic disorder elicited by gluten, occurring in genetically susceptible individuals, characterized by various intestinal and extraintestinal symptoms, the presence of CD-specific antibodies, and enteropathy [1]. Gluten is the common term for peptides found in wheat (gliadin), rye (secalin), and barley (hordein) [2]. Oral intake of gluten in susceptible individuals leads to an inflammatory cascade, resulting in changes in the small intestine, secondarily disrupting nutrient absorption, and causing malabsorption. The global prevalence of CD is around 1%, and 0.8% in the European population [3].

A lifelong strict gluten-free diet (GFD) is the only efficient therapy for CD. The GFD improves health, leads to recovery, and counteracts the adverse effects of CD [4]. A GFD involves the exclusion of wheat, rye, and barley from the diet, together with other products that contain these grains (bread, pasta, flour, and others). As these are the primary sources of energy and nutrients in healthy individuals, similar to other restrictive diets, the GFD poses a higher risk of nutritional inadequacy, especially in children and adolescents, who are in the acceleration phase of linear growth [5].

Regular follow-up with both a physician and a dietitian is a prerequisite for successful compliance to a gluten-free diet, as well as normalization of laboratory tests. With about 52% of paediatric CD patients experiencing problems with dietary adherence, healthcare professionals must regularly monitor and advise patients about the benefits of following a GFD [6]. It has been established that an understanding of CD is likely to improve dietary adherence. Therefore, healthcare professionals should educate and guide patients from the time of diagnosis and through follow-up, especially since patients and their parents are exposed to a plethora of information on the internet [7]. Importantly, patients should be encouraged to seek out the support of patients’ associations, which provide practical information regarding the GFD, especially since membership in CD support groups has been linked to higher dietary adherence [8].

Previous studies regarding the nutritional intake of CD patients have generally shown mixed results. It seems that both healthy children and children on a GFD are at risk of excess fat consumption and insufficient consumption of fibre, iron, vitamin D, and calcium [9]. However, extensive studies in children are still lacking, and therefore, data concerning potential nutritional imbalances remain controversial.

Since there are no existing data concerning the nutritional adequacy of a GFD in Croatia, the aim of the present study was to assess energy intake, nutrient intakes, and anthropometric data of paediatric coeliac patients living on a GFD and compare it to the intake of healthy children on a gluten-containing diet.

## 2. Materials and Methods

Children who were diagnosed with CD (according to the criteria from the ESPGHAN [1]) and regularly followed-up at the Children’s Hospital Zagreb in the period from the year 2014 until 2021, were invited to participate in the study. Participants were asked to keep a 3-day food record (3DFR). Anthropometric measures of CD patients were derived from the last visit to the clinic in those patients who had attended the clinic less than three months before inclusion in the study. When anthropometric data were not available, body weight (BW) and body height (BH) were self-assessed by the parent or caregiver. Anthropometric data were also assessed for each child during the time of diagnosis. The healthy controls (HC) sample included healthy children from a more extensive national database. HC participants were randomly recruited children and adolescents (6 to 14 years old) from selected elementary schools located in continental (Zagreb and Đakovo) and Mediterranean (Pula, Trogir, and Dubrovnik) parts of Croatia. HC participants were recruited in 2019 and 2020. In total, 1000 questionnaires were randomly distributed to children and adolescents by the schools’ principals. Of 1000 distributed questionnaires, 590 complete and correctly filled out questionnaires of children were collected and analysed. In these children, anthropometric data were self-assessed by the caregivers and 3DFRs were filled out either by adolescents themselves or by the caregiver, depending on the age of the child.

Exclusion criteria for patients with CD included a diagnosis of inflammatory bowel disease, type 1 diabetes, or mental health disorders. Healthy controls diagnosed with a chronic disease or who had symptoms of digestive diseases were excluded.

Body mass index (BMI), gender- and age-specific z-scores for height, weight, and BMI were calculated for each child using the PediTools Electronic Growth Chart Calculator [10] with implemented CDC (Centers for Disease Control and Prevention) reference tables [11].

Correct and complete 3DFRs were used for the analysis. The 3DFRs recorded all foods and drinks consumed during the three non-consecutive days, including one day of the weekend. Parents and children were instructed to use kitchen scales to precisely measure intake. Household measures (tablespoons, teaspoons, and cups) were used when that was not possible. Energy and nutrient intake estimated by the 3DFR were analysed using PRODI expert 6.9 software [12]. The intake of each nutrient for every participant was compared to the recommended intakes according to the reference values by the nutrition societies of Germany, Austria, and Switzerland (D-A-CH reference values) [13], and percentages of recommended intakes were evaluated for each participant.

### Statistical Analysis

Continuous variables are presented as mean ± SD and categorical variables as absolute frequencies. Student’s *t*-test was applied to evaluate differences in mean values between CD patients and healthy controls. Differences between categorical data were analysed using the chi-square test. Differences in anthropometric data taken at two different time points for CD patients were analysed using a paired-samples *t*-test. Differences between different groups of CD patients were analysed by the Mann–Whitney U test for nonparametric data. Correlations between certain variables were examined using Pearson’s coefficients. Statistical analysis was performed using SPSS 26.0 (IBM Corporation, Chicago, Illinois, USA) statistical software.

## 3. Results

In total, 163 patients with CD and their parents were asked to participate in the study, of which 128 accepted. Of those, 76 patients with CD (57% female; mean age 9.0 ± 4.3 years) correctly filled-out 3DFRs and were included in the final analysis. Additionally, 590 healthy controls (54% female; mean age 9.9 ± 0.1 years) who correctly and fully completed the 3DFRs were included in the study.

Three participants with CD were not compliant with GFD; 42.1% of participants took oral nutritional supplements at the start of their GFD.

General characteristics and anthropometric data of the whole study sample are summarized in Table 1. In CD patients, comorbidities were present in 16 children (21.1%) and included Gilbert’s syndrome, Hashimoto’s thyroiditis, asthma, allergic rhinitis, vitiligo, headache, and fibromyalgia. Fifty-two (65.8%) patients with CD were from the urban parts of Croatia, while the rest were from the rural parts of Croatia.

Table 2 shows a comparison of energy and nutritional intake between CD patients and healthy controls. There was a statistically significant difference in the intake of most nutrients between the two study groups. Overall, there was better compliance with dietary recommendations in CD patients on the GFD diet compared to healthy children. Inadequate intake of fibre and vitamin D was evident for both CD patients and healthy controls.

There was no statistically significant difference in energy and nutrient intake between CD patients who were on a GFD for 4 years or longer compared to those who were on a GFD for less than 4 years, except for their protein intake (median % of recommended intake 270.4% (63.9, 494.2) in patients who were on a GFD for less than 4 years and 230.3% (99.2, 348.7) for CD patients who were on a GFD for 4 years or longer; *p* = 0.046).

CD patients who were younger than 13 years were significantly more compliant with recommended energy intakes than those who were 13 years and older (median % of recommended intake 98.5% (56.5, 173.2) vs. 78.6% (32.4, 125.6), respectively; *p* = 0.012), dietary fibre (median % of recommended intake 63.2% (13.4, 144.4), vs. 49.9% (25.0, 148.1), respectively; *p* = 0.024), protein (median % of recommended intake 277.5% (99.2, 494.2) vs. 150.7% (63.9, 230.3), respectively; *p* < 0.001), calcium (median % of recommended intake 85.1% (19.2, 221.0) vs. 56.2% (20.4, 80.1), respectively; *p* = 0.002), zinc (median % of recommended intake 124.7% (55.0, 351.0) vs. 76.8% (29.8, 117.6), respectively; *p* < 0.001), magnesium (median % of recommended intake 121.1% (48.8, 416.3) vs. 60.8% (28.1, 109.1), respectively; *p* < 0.001), vitamin C (median % of recommended intake 213.6% (44.7, 705.6) vs. 72.9% (31.3, 330.6), respectively; *p* < 0.001), and vitamin B12 (median % of recommended intake 142.9% (30.5, 759.5) vs. 96.1% (20.8, 162.3), respectively; *p* = 0.018).

Overall, CD patients had significantly lower BW and BH z-scores compared to healthy controls (mean BW z-score 0.2 ± 1.0 in CD patients vs. 0.8 ± 0.9 in healthy controls; *p* < 0.001; mean BH z-score 0.3 ± 1.1 in CD patients vs. 1.3 ± 1.1 in healthy controls; *p* < 0.001). There was no statistically significant difference in BMI z-scores (mean BMI z-score 0.04 ± 0.99 in CD patients vs. 0.22 ± 1.06 in healthy controls; *p* = 0.186).

For CD patients, BW, BH, and BMI z-scores from the time of diagnosis were compared with scores from the time when 3DFRs were collected. Mean BW, BH, and BMI z-scores in CD patients increased significantly after the mean duration of 34.1 months on a GFD (mean BW z-score −0.4 ± 1.3 before GFD to 0.2 ± 1.0 after GFD, *p* < 0.001; mean BH z-score 0.2 ± 1.0 before GFD to 0.3 ± 1.1 after GFD; *p*= 0.007; mean BMI z-score −0.3 ± 1.2 before GFD to 0.0 ± 1.0 after GFD; *p* = 0.012).There was no correlation between the duration of the GFD and mean BW, BH, and BMI z-score.

Figure 1 shows the distribution of underweight, normal weight, and overweight between CD patients at the time of diagnosis, CD patients on a gluten-free diet, and healthy controls. There was a rise in overweight in participants with CD after the introduction of a GFD, although this was not statistically significant (11.1% before GFD, 17.8% after GFD; *p* = 0.432).

## 4. Discussion

In our study, patients with CD on a GFD were more compliant with dietary recommendations than healthy controls and had an overall nutritional intake that was more in accordance with dietary recommendations. This was especially evident for younger patients, while in adolescents, a deviation from recommended intakes, similar to that observed in their healthy counterparts, was evident. Although the nutritional status of CD patients improved while they were on a GFD compared to at the time of diagnosis, mean BW and BH z-scores were still significantly lower than those of healthy controls.

To date, a number of studies have investigated differences in the nutritional intake of patients with CD in comparison to healthy controls [14,15,16,17,18,19,20,21,22,23]. The results of those studies were mixed, mainly focused on the intake of particular nutrients rather than the diet as a whole. Overall, the nutritional intake of children with CD consistently showed a lower intake of fibre [19,24] and a higher intake of fat [14,15,17] compared to healthy controls. However, in most of the studies, the overall dietary intake of CD patients was generally similar to that of healthy controls commonly not meeting the recommended intakes for iron, calcium, vitamin D, and fibre for both groups. Nevertheless, some studies have found that both patients with CD and healthy controls had adequate micronutrient intakes [17,21]. Furthermore, a study by Forchielli et al. [25] showed that the nutritional intake of children on one year of a GFD is nutritionally very similar to their diet before the introduction of GFD, with some improvements, especially regarding unsaturated fat and fibre intake. Similar to previous studies, in our study, patients with CD had a low intake of fibre and vitamin D. However, their intake was not statistically different from healthy controls. Surprisingly, our study found that the nutritional intake of patients with CD was overall compliant with dietary recommendations, whereas healthy controls deviated from recommendations for the majority of macro- and micronutrients. However, our sample consisted mainly of younger children up to 12 years old (*N* = 62, 81.6%), whose dietary intakes are still strongly influenced by their parents. It should be noted that all of our patients and their parents received nutritional education from our department dietitians at the start of their GFD regime and were thereafter followed up every 3 to 6 months. Therefore, we assume parents of younger children put greater focus and time on their children’s diet than parents of healthy children. In contrast, adolescents’ dietary intake in our sample tended to be less balanced and more similar to that of healthy adolescents’ diets.

We also assessed the nutritional status of children with CD. Studies have shown a positive correlation between adherence to the GFD and growth percentiles, which increases when higher adherence scores are achieved [26]. Although the nutritional status of children with CD in our sample improved significantly while they were on a GFD, it did not completely reverse and catch up with healthy children despite that majority of our patients were following a strict GFD. A similar finding was observed in a study by Ting et al. [21]. Although only 3 out of 76 patients in our study were not compliant with a GFD, it is likely that majority of patients who were not strictly avoiding gluten did not accept participation in the study. Indeed, Jadrešin et al. [27] have found that only 59% of Croatian patients aged between 5 and 30 years follow a strict GFD.

Our study has some limitations. Firstly, we were not able to personally assess the BW and BH of all CD patients and healthy controls, therefore their BW and BH were self-assessed by their parents, certainly leading to a high risk of errors in measurements. Moreover, we did not match controls by gender and age. However, the age distribution of our samples was similar and no significant difference in age was observed between CD patients and healthy controls. Furthermore, to circumvent possible errors caused by different age distributions, for each child, proportions of recommended nutrient intakes according to their age and gender were assessed, and for anthropometric measures, gender and age-specific z-scores were determined. Nevertheless, it should be noted that children and parents of children with CD who are not compliant with the GFD were probably less likely to accept participation in the study, and therefore, we might have missed a proportion of those patients. As with any dietary method, risk of bias is always present as participants can always either change their behaviour (for example, make healthier food choices) and/or, omit recording “unfavourable” foods in the diary. However, the bias will be present in both patients with CD and healthy controls; therefore, the final error will be minimized. We have not assessed the socioeconomic status of the children/their parents, which can impact the dietary habits of the children involved. It should also be noted that this study is only representative of the population of children and adolescents in Croatia.

## 5. Conclusions

Our study showed that patients with CD have better compliance with dietary recommendations compared to healthy controls, showing that a well-balanced GFD diet can provide necessary macro- and micronutrients. This study confirms the importance of regular dietary consultations with a dietitian specialized for a GFD not only for the need for strict gluten-free avoidance but also to educate parents and their children about a GFD diet that is nutritionally balanced and that provides all nutrients needed for optimal growth and development of the child.

## Figures and Tables

**Figure 1 children-09-01595-f001:**
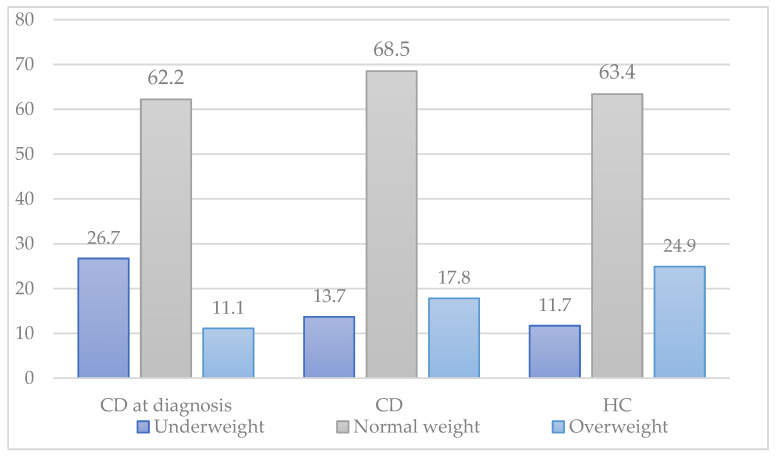
Distribution of underweight, overweight, and normal weight of CD patients at diagnosis (CD at diagnosis), CD patients during GFD (CD), and healthy controls (HC).

**Table 1 children-09-01595-t001:** General characteristics of study samples.

	Patients with Coeliac Disease (*N* = 76)	Healthy Controls (*N* = 590)	*p*-Value
Age (years), mean ± SD	9.0 ± 4.3	9.9 ± 0.1	0.06
Female, *n* (%)	43 (57%)	317 (54%)	0.65
Age at diagnosis (months), mean ± SD	72.8 ± 49.4		
Duration of GFD (months), mean ± SD	34.1 ± 25.4		
Body weight (kg), mean ± SD	33.3 ± 12.8	39.8 ± 12.8	0.002
Body weight z-score, mean ± SD	0.2 ± 1.0	0.8 ± 0.9	<0.001
Body height/length (cm), mean ± SD	133.0 ± 24.7	147.0 ± 14.6	<0.001
Body height/length z-score, mean ± SD	0.3 ± 1.1	1.3 ± 1.1	<0.001
Body mass index (kg/m^2^), mean ± SD	17.6 ± 3.3	18.1 ± 3.2	0.27
Body mass index z-score, mean ± SD	0.0 ± 1.0	0.2 ± 1.1	0.186

**Table 2 children-09-01595-t002:** Comparison of nutritional intake between CD patients on GFD and healthy controls.

	CD Patients (*N* = 76)	Healthy Controls (*N* = 590)	*p*-Value
Energy (kcal), mean ± SD	1740.3 ± 482.2	1454.8 ± 423.9	<0.001
Energy requirements (%), mean ± SD	96.4 ± 25.7	72.7 ± 22.7	<0.001
Dietary fibre (g), mean ± SD	12.9 ± 5.1	13.2 ± 6.6	0.739
Dietary fibre requirements (%), mean ± SD	64.5 ± 26.1	61.5 ± 33.4	0.462
Total fat (g), mean ± SD	70.0 ± 26,7	52.4 ± 19.7	<0.001
Total fat requirements (%), mean ± SD	35.8 ± 7.0	32.3 ± 6.3	<0.001
Monounsaturated fatty acids requirements (%), mean ± SD	8.4 ± 2.9	9.5 ± 3.4	0.003
Polyunsaturated fatty acids requirements (%), mean ± SD	4.1 ± 2.1	4.0 ± 2.0	0.663
Saturated fatty acids requirements (%), mean ± SD	11.6 ± 3.7	12.0 ± 3.5	0.399
Protein (g), mean ± SD	67.0 ± 19.4	62.1 ± 17.1	0.021
Protein requirements (%), mean ± SD	262.5 ± 108.3	190.3 ± 79.1	<0.001
Calcium (mg), mean ± SD	713.3 ± 261.2	579.9 ± 242.1	<0.001
Calcium requirements (%), mean ± SD	81.8 ± 37.6	57.7 ± 25.9	<0.001
Iron (mg), mean ± SD	8.2 ± 2.9	8.0 ± 4.3	0.584
Iron requirements (%), mean ± SD	80.7 ± 30.3	69.1 ± 42.5	0.021
Zinc (mg), mean ± SD	7.4 ± 2.5	8.0 ± 2.8	0.077
Zinc requirements (%), mean ± SD	130.7 ± 67.2	109.6 ± 48.9	0.010
Vitamin B1 (mg), mean ± SD	1.5 ± 1.2	1.7 ± 1.5	0.145
Vitamin B1 requirements (%), mean ± SD	176.3 ± 137.4	185.9 ± 162.5	0.578
Vitamin C (mg), mean ± SD	91.9 ± 51.8	74.1 ± 49.8	0.004
Vitamin C requirements (%), mean ± SD	230.9 ± 158.8	136.2 ± 103.9	<0.001
Vitamin D (µg), mean ± SD	2.2 ± 2.4	1.1 ± 1.7	<0.001
Vitamin D requirements (%), mean ± SD	10.9 ± 11.9	5.2 ± 8.3	<0.001

## Data Availability

All data supporting the reported results are deposited at the Children’s Hospital Zagreb in written and electronic format.

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
