# Peer review of "Quality of Diet of Patients with Coeliac Disease in Comparison to Healthy Children"

_children, 2022, doi:10.3390/children9101595_

Round 1
Reviewer 1 Report
Hello,
The manuscript titled “Quality of diet of patients with coeliac disease in comparison to healthy children” compares the diet intake between coeliac vs healthy children and their adherence to the diet plan. The present study outlines the younger children with coeliac disease are likely to follow the routine and diet regime. The strengthens the study, the authors should consider below points.
Major:
1. The study should investigate the physical activity record in addition to the BM and BH charts.
2. Also, family status plays bigger determinants for children to follow the certain routines. The size of the family, economic status and education background can impact the adherence of certain regime in children.
3. There should be component to add for discussion whether children with or without CD have other underlying systemic or immunological disorder.
4. Authors should include more literature in the area and discuss what has been effective in past in CD patients. Discuss some supportive programs or groups to educate the patients and how this has been benefited or nor to the community.
Minor:
1. In abstract, line 16,17 and 18 requires re writing with omitting the repeating sentence.
2. BW and BH acronym need full form in line 64 in addition to the abstract.
3. Line 82, there is error, the title requires the numbering.
4. Line 79, what is D-A-CH. Needs explanation.
5. Line 229, typo error at the start.
6. Requires, proofread for typo error. On many occasions, the wrong/missed capitalization of letter observed.
Author Response
Dear reviewer,
We send our revised manuscript entitled “Quality of diet of patients with coeliac disease in comparison to healthy children” for possible publication in the journal Children. We would like to thank you for your useful comments which are all addressed in detail below. We hope that you will find revised manuscript suitable for publication in the journal.
Reply to the comments:
Hello,
The manuscript titled “Quality of diet of patients with coeliac disease in comparison to healthy children” compares the diet intake between coeliac vs healthy children and their adherence to the diet plan. The present study outlines the younger children with coeliac disease are likely to follow the routine and diet regime. The strengthens the study, the authors should consider below points.
Major:
- The study should investigate the physical activity record in addition to the BM and BH charts.
Authors: Thank you for your comment. Although we do agree that data on physical activity is important in these types of studies, because of a very large sample size and methodological difficulties related to collection of high quality data on physical activity (requirement of accelerometers to precisely assess physical activity), we have decided it will not be collected at this instance. Moreover, this was out of the scope of our study aim, as our main goal was to determine quality of diet of children on GFD.
- Also, family status plays bigger determinants for children to follow the certain routines. The size of the family, economic status and education background can impact the adherence of certain regime in children.
Authors: Thank you very much for your useful comment. We agree that family plays a vital role in both provision of regular meals and food, and more importantly, determines dietary habits of both healthy children and children with coeliac disease. Although this data would add on to the data we do have, we deem it is not a requisite for our study, since we have a very large control group of healthy children. Therefore, any bias that might have been caused by the differences in socioeconomic status of children with CD was avoided. However, we have added this as the limitation of our study in the Discussion section of our manuscript.
- There should be component to add for discussion whether children with or without CD have other underlying systemic or immunological disorder.
Authors: Thank you very much for your comment. Although other systemic or immunologic disorders can indeed be present in children with CD, we think this does not affect the results of our study (nutritional intake of children with CD). Therefore, if reviewer and editor agree, we would like to try to keep the article concise and simple, and leave out the requested section out of the manuscript.
- Authors should include more literature in the area and discuss what has been effective in past in CD patients. Discuss some supportive programs or groups to educate the patients and how this has been benefited or nor to the community.
Authors: Thank you very much for you useful comment. We agree with your proposal. Paragraph regarding different programs and importance of regular follow-ups of patients with CD has been added to the introduction section of our manuscript.
Minor:
- In abstract, line 16,17 and 18 requires re writing with omitting the repeating sentence.
Authors: Thank you for your comment. The sentences you are mentioning goes as follows: „In all participants anthropometric data were determined at the time when 3DFRs were collected. In CD patients anthropometric data were also determined at diagnosis.“ As this seems to be confusing, we have changed it into: „In healthy controls anthropometric data were determined at the time when 3DFRs were collected. In CD patients anthropometric data were determined at two time points: at diagnosis and at the time of 3DFRs collection.“
- BW and BH acronym need full form in line 64 in addition to the abstract.
Authors: Thank you for your comment. We have done as requested.
- Line 82, there is error, the title requires the numbering.
Authors: Thank you for your comment. We have done as suggested.
- Line 79, what is D-A-CH. Needs explanation.
Authors: Thank you for your comment. We have done as suggested.
- Line 229, typo error at the start.
Authors: Thank you for your comment. We have done as suggested.
- Requires, proofread for typo error. On many occasions, the wrong/missed capitalization of letter observed.
Authors: Thank you for your comment. We have done as suggested.
Reviewer 2 Report
The authors have shown interesting results but I found several weaknesses in the study, especially in the study methodology. My comments are below:
Comment#1: The method section contains the results of the study that must be shifted into the result section.
Comment#2: The study methodology is not transparent. I am confused about understanding the participants' recruiting process.
Comment#3: If the authors wanted to compare the nutritional deficiency in healthy and CD children. Why did the authors include their patients for 3DFR? Moreover, in the result section, there is no result about the parents.
Comment#4: The authors did not explain the inclusion/exclusion criteria.
Comment#5: They explained how many parents were contacted and how many responses but they did not explain about CD children and healthy participants in the same way.
Comment#6: Seventy-six patience is a less number to draw confusion. There is a huge gap between the CD (n=76) and healthy children (590). The authors did not explain this.
Comment#7: If patients know their food records will be analyzed, apparently the food selection will be biased. Patients will not choose under-nutritious food.
Comment#8: The authors also included food consumed on the weekends but there is no information if there was any difference in weekend and weekdays food.
Comment#9: There were CD patients with more or less than 4 years of GFD but the actual comparison was based on 3 days of food record which is a highly insufficient time to draw a significant conclusion.
If these points are correct and if I understood well. The result of the study is weak and cannot be reliable.
Comment#10: Nutrition is dependent on several factors (type of food, financial situation, gluten contamination) if the study was conducted in Croatia in Croatian children. The result cannot be compared to the developing/underdeveloped countries. Hence, the authors should mention the study represent the situation in Croatia only.
Comment#11: What is PRODI expert? Line no. 19
Comment#12: What are BH and BW? Line no. 64
Comment#13: Please mention the duration of the study.

Author Response
Dear reviewer,
We send our revised manuscript entitled “Quality of diet of patients with coeliac disease in comparison to healthy children” for possible publication in the journal Children. We would like to thank you for your useful comments which are all addressed in detail below. We hope that you will find revised manuscript suitable for publication in the journal.
Reply to the reviewer:
The authors have shown interesting results but I found several weaknesses in the study, especially in the study methodology. My comments are below:
Comment#1: The method section contains the results of the study that must be shifted into the result section.
Authors: Thank you very much for you useful comment. We have done as suggested.
Comment#2: The study methodology is not transparent. I am confused about understanding the participants' recruiting process.
Authors: Thank you very much for you useful comment. We have explained recruiting process in more details, as suggested.
Comment#3: If the authors wanted to compare the nutritional deficiency in healthy and CD children. Why did the authors include their patients for 3DFR? Moreover, in the result section, there is no result about the parents.
Authors: Thank you for your comment and question. We are sorry for the misunderstanding. Parents of included children were not included as participants. Parents were only instructed to help collect the data on anthropometric measures and food intake of young children who are not able to do it themselves. We have now explained it more precisely in the method section of our manuscript.
Comment#4: The authors did not explain the inclusion/exclusion criteria.
Authors: Thank you for your comment. Indeed, this has now been added to the Method section of our manuscript.
Comment#5: They explained how many parents were contacted and how many responses but they did not explain about CD children and healthy participants in the same way.
Authors: Thank you very much for you useful comment. We have explained recruiting process in more details, as suggested in a Comment#2..
Comment#6: Seventy-six patience is a less number to draw confusion. There is a huge gap between the CD (n=76) and healthy children (590). The authors did not explain this.
Authors: Thank you for your comment. The prevalence of coeliac disease is about 1%. In our centre, yearly about 12 (in 2014) up to 37 (2021) children are diagnosed with CD. Moreover, usage of 3-day food diaries (3DFRs) is considered gold standard in nutritional research – however, it is very demanding and timely for the participant to collect it – therefore, final number of children with CD included in the study is, in our opinion, adequate and comparable or larger than in other studies. As for healthy controls, for the sample to be representative of healthy children from Croatia (and, therefore, comparable to the sample of children with CD), large number of children of different age from different regions of Croatia was required.
Comment#7: If patients know their food records will be analyzed, apparently the food selection will be biased. Patients will not choose under-nutritious food.
Authors: Thank you for your comment. In nutritional studies, the issue of bias in food intake data is always present. Unfortunately, no perfect methodological tool to determine dietary intake exists up to this date, and all available tools (food records, food frequency questionnaire, 24-h dietary recall) have some strengths and limitation (please refer to the following the papers: https://pubmed.ncbi.nlm.nih.gov/8279436/, https://www.ncbi.nlm.nih.gov/pmc/articles/PMC8877528/) . However, food diary is considered gold standard for determining the nutritional intake in research as it provides the most detailed overview of dietary intake during 3 days, which is representative for usual intake. As with any other type of questionnaire, the participant can always either change his behaviour (in this case, as you said, make healthier food choices) and/or, exclude “unfavourable” foods from writing in the diary. However, the bias will be present in both patients with CD and healthy controls, therefore the final error will be minimal.
Comment#8: The authors also included food consumed on the weekends but there is no information if there was any difference in weekend and weekdays food.
Authors: Thank you for your comment. Protocol for filling-out 3-day food record instructs respondents to write their diaries for 3 non-consecutive days, of which one should be the day of the weekend (which is expected to be different than the weekday). When food diaries are analysed, the average intake over 3 days is calculated, as one day is not representative for the usual intake. It is not a practice in this type of research to analyse weekend and weekdays separately, nor was this the aim of our research.
Comment#9: There were CD patients with more or less than 4 years of GFD but the actual comparison was based on 3 days of food record which is a highly insufficient time to draw a significant conclusion.
If these points are correct and if I understood well. The result of the study is weak and cannot be reliable.
Authors: Thank you for your comment. Unfortunately we cannot agree with the points you are mentioning. Indeed, as you can see in papers, dietary intake cannot be estimated without errors (https://pubmed.ncbi.nlm.nih.gov/8279436/, https://www.ncbi.nlm.nih.gov/pmc/articles/PMC8877528/). However, 3-day food records are considered the most accurate tools for analysing usual food intake in nutritional studies. This study is a cross-sectional study, meaning it is an observational research that analyses data of variables collected at one given point in time. The aim of this study was not to observe the change in dietary intake of children with CD over time – that would require different methodology, and using multiple 3-day food diaries. But rather, as we have explained, the aim of this study was to give a “time-shot” of usual dietary intake in children with CD, who were on average on gluten-free diet for 2.8 years (34.1 months) (some less, some more) and to compare it to healthy children. Furtheremore, we have analyzed difference in nutritional intake of children who were on a GFD for 4 and more years, as compared to those who were on a GFD less than 4 years, and quoting from the manuscript: “There was no statistically significant difference in energy and nutrient intake between CD patients who were on a GFD for 4 and more years compared to those who were on a GFD less than 4 years, except for the intake of proteins (median % of recommended intake 270.4% (63.9, 494.2) in patients who were on a GFD for less than 4 years and 230.3% (99.2, 348.7) for CD patients who were on a GFD for 4 years or longer; p=0.046).” We kindly suggest to you to look at other similar studies cited in our manuscript.
Comment#10: Nutrition is dependent on several factors (type of food, financial situation, gluten contamination) if the study was conducted in Croatia in Croatian children. The result cannot be compared to the developing/underdeveloped countries. Hence, the authors should mention the study represent the situation in Croatia only.
Author: Thank you for your comment. We accept your suggestion and have added this into study limitations.
Comment#11: What is PRODI expert? Line no. 19
Author: Thank you for your comment. PRODI expert is a software used for the analysis of nutritional intake, commonly used in nutritional research.
Comment#12: What are BH and BW? Line no. 64
Author: Thank you for your comment. BH and BW are abbreviations which have now been explained in the Manuscript.
Comment#13: Please mention the duration of the study.
Author: Thank you for your comment. The duration of the study has been added into the Methodology section of the Manuscript.
Round 2
Reviewer 1 Report
Hello,
Thank you for providing the justification to the reviewers comments. Please see below comments for further improvements.
1. The scope of the paper is very narrow and requires more factors to be included. Please consider the broadening the questionnaires about suggested points such as economic data, geographic data and co-morbid immuno compromised condition. The manuscript lacks important aspects of Quality diet and its relation to other aspects and how it interplays. Adding this section or adding related literature the opens quality of paper and interests more readers.
2. The study can be small size population for healthy control. More importantly, the CD patient sample size is not huge so please consider above points.
3. Please also consider adding the more assessment and points for diet in children vs young adults. Merely, BW and BH data is not enough the project the future trend. It would be wise for author to reconsider scope of the paper.
Author Response
Dear Reviewer,
We send our revised manuscript entitled “Quality of diet of patients with coeliac disease in comparison to healthy children” for possible publication in the journal Children. We would like to thank you for your useful comments which are all addressed in detail below. We hope that you will find revised manuscript suitable for publication.
Reviewers comments:
Hello,
Thank you for providing the justification to the reviewers comments. Please see below comments for further improvements.
- The scope of the paper is very narrow and requires more factors to be included. Please consider the broadening the questionnaires about suggested points such as economic data, geographic data and co-morbid immuno compromised condition. The manuscript lacks important aspects of Quality diet and its relation to other aspects and how it interplays. Adding this section or adding related literature the opens quality of paper and interests more readers.
Authors: Thank you for your comment. As suggested, we have added additional data regarding geographic data and comorbidities present in patients with CD on a GFD, which we believe will add to the quality of the manuscript. Unfortunately, as a large number of healthy controls were included into this study, and since healthy controls are blinded to the researcher regarding their identity, we are not able to acquire additional data for healthy controls. However, we do believe that said data is not requisite for this study’s purpose.
- The study can be small size population for healthy control. More importantly, the CD patient sample size is not huge so please consider above points.
Authors: Thank you for your comment. As explained in the previous comment, we have added available data to the manuscript.
- Please also consider adding the more assessment and points for diet in children vs young adults. Merely, BW and BH data is not enough the project the future trend. It would be wise for author to reconsider scope of the paper.
Authors: Thank you for your comment. Unfortunately, as you are surely aware, it is not possible to retroactively collect additional data for healthy controls , as all data collected in healthy controls was performed according to the ethical study procedures, meaning all healthy controls were blinded to the researcher. Moreover, this study is a cross-sectional study, meaning it is an observational research that analyses data of variables collected at one given point in time. The aim of this study was not to observe the change in dietary intake or anthropometric measures in children with CD over time – that would require different methodology, and using multiple measurements. But rather, the aim of this study was to give a “time-shot” of usual dietary intake and nutritional status of children WHILE being on a gluten-free diet. Anthropometric measures of children with CD measured before the introduction of GFD add to the existing data, and give readers better “compass” of the appropriateness of such dietary regimen.
Reviewer 2 Report
I am happy with the most of the response provided by the authors. They have done a great job. However, I still have some suggestions.
Comment#1: Line no. 94-98 looks like a part of line no 75-76. Some lines are overlapping; I suggest clubbing both the lines together.
Comment#2: Please define the healthy controls? Line no. 86-91
Comment#3: I am happy with the responses for the comment no 6 and 7 (V1). These are common confusions. I suggest mentioning these in the discussion part.
Comment#4: Please provide the full form of the HC. Line no. 86
Author Response
Dear Reviewer,
We send our revised manuscript entitled “Quality of diet of patients with coeliac disease in comparison to healthy children” for possible publication in the journal Children. We would like to thank you for your useful comments which are all addressed in detail below. We hope that you will find revised manuscript suitable for publication.
Reviewer 2, round 2
I am happy with the most of the response provided by the authors. They have done a great job. However, I still have some suggestions.
Comment#1: Line no. 94-98 looks like a part of line no 75-76. Some lines are overlapping; I suggest clubbing both the lines together.
Authors: Thank you for your comment. We have done corrections accordingly.
Comment#2: Please define the healthy controls? Line no. 86-91
Authors: Thank you for your comment. We have added more details regarding healthy controls.
Comment#3: I am happy with the responses for the comment no 6 and 7 (V1). These are common confusions. I suggest mentioning these in the discussion part.
Authors: Thank you for your comment. We have added this in the discussion part (limitations) as suggested.
Comment#4: Please provide the full form of the HC. Line no. 86
Authors: Thank you for your comment. We have done corrections accordingly.
Round 3
Reviewer 1 Report
Hello,
Thank you for providing the updated version.
Author Response
Thank you for your suggestions once again.
Best regards.